# Experimental Medicine for HIV Vaccine Research and Development

**DOI:** 10.3390/vaccines11050970

**Published:** 2023-05-11

**Authors:** Holly Prudden, Roger Tatoud, Cathy Slack, Robin Shattock, Pervin Anklesaria, Linda-Gail Bekker, Susan Buchbinder

**Affiliations:** 1Independent Consultant, London TN7 4AS, UK; 2Origena Consulting, 01210 Ferney Voltaire, France; r.tatoud@outlook.com; 3HIV/AIDS Vaccines Ethics Group (HAVEG), University of KwaZulu-Natal, Durban 4041, South Africa; slackca@ukzn.ac.za; 4Imperial College London, London SW7 2BX, UK; r.shattock@imperial.ac.uk; 5Bill & Melinda Gates Foundation, Seattle, WA 98109, USA; pervin.anklesaria@gatesfoundation.org; 6The Desmond Tutu HIV Centre, Faculty of Health Sciences, University of Cape Town, Cape Town 7925, South Africa; linda-gail.bekker@hiv-research.org.za; 7Bridge HIV, San Francisco Department of Public Health, San Francisco, CA 94102, USA; susan.buchbinder@sfdph.org; 8Department of Epidemiology and Biostatistics, University of California, San Francisco, CA 94158, USA

**Keywords:** clinical trial, HIV vaccine, HIV prevention, experimental medicine, research methods

## Abstract

The development of safe and effective HIV vaccines has been a scientific challenge for more than 40 years. Despite disappointing results from efficacy clinical trials, much has been learnt from years of research and development. In a rapidly evolving HIV prevention landscape, swift evaluation of multiple vaccine approaches eliciting cross-reactive humoral and cellular responses is needed to ensure the development of efficacious vaccine candidates. To contain increasing costs, innovative clinical research methods are required. Experimental medicine has the potential to accelerate vaccine discovery by iterating early stages of clinical testing faster and by selecting the most promising immunogen combinations for further clinical evaluation. As part of its mission to unite diverse stakeholders involved in the response to the HIV epidemic, the Global HIV Vaccine Enterprise at IAS—the International AIDS Society—hosted a series of online events between January and September 2022 to discuss the merits and challenges of experimental medicine studies to accelerate the development of safe and effective HIV vaccines. This report summarizes key questions and discussions across the series of events, which brought together scientists, policy makers, community stakeholders, advocates, bioethicists, and funders.

## 1. Introduction

There has been significant progress in HIV prevention due to the recognition of secondary prevention of onward transmission benefits of undetectable viral load in people on effective antiviral therapy and the advent of highly effective antiretroviral (ARV)-based HIV prevention in the form of pre- and post-exposure prophylaxis. However, 1.5 million people acquired HIV in 2021, adding to the 38.4 million people globally living with the virus [1]. Treatment targets set up for 2020 were missed, the 95-95-95 targets for 2025 are still to be attained, and ARV-based pre-exposure prophylaxis remains inaccessible to many of those in need of HIV prevention tools.

There are compelling reasons to believe that primary prevention is key, and that a highly efficacious and durable preventive HIV vaccine will permit robust control of the epidemic [2]. However, an HIV vaccine remains elusive. The disappointing results of the HVTN 702/Uhambo, HVTN 705/Imbokodo, and HVTN706/Mosaico large-scale efficacy trials [3,4] have prompted the HIV vaccine research and development (R&D) field to consider new approaches to developing vaccines, especially those that can elicit cross-reactive and potent neutralizing antibodies and cellular (CD8 T cell) responses and new ways of rapidly testing multiple immunogens in clinical studies to accelerate the development of an HIV vaccine.

Experimental medicine (EM) is an attractive approach allowing rapid hypothesis testing to support and accelerate the development of better vaccine candidates. In this report, we define EM for HIV vaccine R&D as “clinical investigations of multiple immunogens undertaken to test or generate a scientific *hypothesis* that advances vaccine discovery and development rather than testing a specific *product* being designed to move into later-phase trials”.

To explore the merits and challenges of EM for HIV vaccine R&D, the Global HIV Vaccine Enterprise at IAS—the International AIDS Society—organized a series of webinars bringing together scientists, policy makers, advocates, community stakeholders, funders, and bioethicists involved in the HIV response (Table 1). This report summarizes the presentation and discussions from this series of events.

## 2. The Merits of EM to Accelerate HIV Vaccine Research and Development

It was noted that safe, effective HIV vaccines remain a necessity for the durable control of the epidemic. There is a strong public health need to accelerate vaccine development, especially those that can elicit a comprehensive and robust immune response, including cross-reactive neutralizing antibodies. It is understood from natural infection that multiple immunogens given sequentially are potentially required to generate cross-reactive neutralizing responses. Thus, to accelerate the pace of immunogen selection and develop an optimal regimen, multiple small-scale, hypothesis-testing clinical studies are necessary. Conventional vaccine development approaches are thus not suited for identification and selection of a regimen that will elicit potent neutralizing and cellular responses. Once a regimen of sequential immunogens is identified, the more traditional vaccine testing in Phase 2 efficacy study can be implemented. Such an approach will mitigate the risk of testing unfavorable regimens in expensive Phase 2 studies. Given the expected complex prevention landscape, Phase 2 clinical studies will require careful consideration for inclusion of placebo controls and efficacy demonstration may require novel study designs.

It was proposed that it is no longer justifiable to propose testing new candidates through expensive Phase 2 and 3 trials when resources could be allocated to increasing the likelihood of success by first using approaches such as EM as a tool to design products with a higher chance of success. Further, effective HIV prevention tools (mainly pre-exposure prophylaxis) create new challenges for the design and conduct of efficacy studies [5]. Identifying new strategies to expedite these earlier experimental stages to identify and select immunogens in a faster iterative fashion is now required. In parallel, there is a need to continue to increase our understanding of the immunological rules that drive cross-reactive, potent, and durable humoral and cellular immunity against key HIV epitopes.

It was noted that the aim of EM studies is to accelerate vaccine science, without compromising participant safety or clinical research standards, through iterative research with little or no expectation that individual immunogens will progress further in their own right; they may be part of an eventual vaccine regimen (Box 1). Commentators also suggested that EM is not a new concept and is considered by some as similar to Phase 0 or first-in-man studies [6]. Experimental medicine studies have been extended to the field of vaccines [7], and Table 2 shows a summary of key differences between EM and Phase 1 trials.

Box 1The EM approach.
EM trials are designed to accelerate HIV vaccine development, increasing the probability of success for products moving into clinical evaluation.Address questions that are not capable of definitive solutions solely in animal models—humans are the best model for humans.Provide opportunities for early iterations between preclinical and clinical research (para-clinical approach).Early validation and sequential iterations for structural-based design.Evaluate novel concepts prior to formal product development—hypothesis testing.Involve in-depth analysis of human specimens, such as next-generation sequencing (NGS) to analyze antibody germ line engagement and evolution.May involve intense sampling, such as mucosal or lymph node biopsies and daily blood samples.


It was suggested that EM trials can provide more pertinent answers to focused questions by allowing for rapid iterations between preclinical and clinical research and can provide opportunities to improve and evaluate novel concepts prior to formal product development. Event participants noted that the decision to proceed with an EM approach should be justified by stringent scientific and ethical rationales, as for any other trials. In this regard, pre-clinical safety assessment in animals will continue to be performed prior to testing in EM trials. Above all, investigators have a duty to ensure that the candidate tested presents an acceptable level of risk to participants and is justified by the benefits, in this case in the form of benefits to society (knowledge gains). Investigators should address the need to accumulate enough scientific evidence to proceed with EM studies, irrespective of whether the candidate might be taken forward or not. Safety considerations remain paramount.

Clinical guidelines (ICH-GCP) and ethics guidelines (UNAIDS and AVAC Good participatory practice Guidelines for biomedical HIV prevention trials [8] and UNAIDS/WHO Ethical considerations in HIV prevention trials [9]) identify value in studies as a scientific and ethical requirement. EM trials are not an exception because one of the great values of clinical research is in the dissemination and sharing of data and results widely (including to communities participating in the research), allowing others to build on the ideas and results, and further respecting and valuing the contribution of participants in these trials.

Participants noted that although EM trials may not ensure direct prophylactic or therapeutic benefit to the study participants from the experimental intervention itself, some benefit may be yielded by other study procedures. These include enhanced personal knowledge of a medical condition, receipt of a sound standard of prevention, and early referral for health conditions identified in the study. EM trials will provide important scientific information to advance product development, which would not be possible without the contribution of volunteers.

## 3. Challenges of Conducting EM Trials

It was noted that EM requires identifying and addressing specific regulatory, ethical, and clinical considerations.


**Several regulatory considerations were discussed**


Currently, there are no specific regulatory guidelines for conducting EM studies. Further, different regulatory bodies follow different rules and levels of flexibility. New legislation or policies should address the need for a regulatory framework for EM and may also signal a move towards achieving a global standard for the conduct of EM trials.


**Several ethics considerations were noted**


Regarding *stakeholder engagement*, Research Ethics Committees (RECs), Community Advisory Boards (CABs), community stakeholders, and participants need to acquire knowledge and appreciation of the nature and primary goal of this type of research, which should be explained in appropriate terms tailored for a variety of stakeholders. The use of highly adaptive trial designs adds to the complexity of the research and to its communication to both regulatory and ethical bodies, in addition to participants. Scientists, CAB members, community stakeholders, REC reviewers and study participants should all have a shared understanding that the goal of the study is to identify products for future efficacy testing. With such trials, there are risks attached to misunderstanding the central purpose of EM studies and, consequently, a greater effort is needed from the research teams to engage community stakeholders to understand the nature and aims of the research, and to solicit their views and inputs. WHO/UNAIDS Good Participatory Practice provides a helpful framework for engagement in such trials.

Regarding *risk–benefit ratios*, the intervention in EM trials will not hold out the prospect of direct benefit for individuals. Therefore, its risks will be judged against benefits to society in terms of knowledge gains. Communicating the risk–benefit ratio of EM trials will inform needed discussions with relevant stakeholders. This is similar to risk–benefit ratios of early-phase clinical research and development. It is important to explain the social value of EM, which typically relates to the issue of relevance to the whole community of stakeholders. It is important to acknowledge the conflict between what is in the best interests of those who participate in a trial and what is in the best interest of future populations and may arise from benefits downstream of the intervention.

Regarding *informed consent*, the goal of EM trials should be highlighted in informed consent forms and consent discussions with study participants. This is to offset any potential misunderstanding that the study intervention may confer direct health-related benefits to participants or even that the study goal is to move products through a pipeline towards introduction in vulnerable communities. EM trials may require recruiting diverse types of participants. As with all vaccine research, providing accessible informed consent and study documentation, in addition to consent discussions, is paramount to successfully engaging potential participants. Particular attention should be given to the informed consent documentation to ensure that it is accessible while respecting cultural specificities and individual autonomy. An innovative design building on visuals is now recognized as a very powerful way to communicate, in addition to text and spoken language. Partnering with CABs is essential to design consent forms and strategies that are relevant to the study and the settings in which it is taking place. While no method of informed consent is perfect, there is a need to put in place mechanisms to gauge the best approaches.

Regarding *payment* to participants, EM raises the question of whether different incentives are required for participation in studies that may hold out higher burdens for participants. There is a consensus that trial participants should be compensated for their time and expenses incurred. To help ensure that payments are fair and not so excessive as to undermine sound decision making, CAB members can advise on the nature and level of compensation. More than ever, volunteers should be considered as “research partners” in EM.


**Several clinical issues were also raised**


The duration of an EM trial is usually shorter than other trials and requires a dedicated and experienced clinical research site with a particular skill set. Extensive sampling can be burdensome to participants and clinical staff. Novel assays may be required, and this may necessitate either building laboratory capacity locally or ensuring efficient and effective shipping of samples. In addition, there are challenges in the manufacturing of trial vaccine candidates. The cost and time required to manufacture a clinical batch can delay EM research and increase costs substantially. Time and cost savings could be achieved through the development of alternative manufacturing and regulatory pathways. It is also conceivable that EM studies could be performed as part of or alongside more advanced trial phases. This is an opportunity to be explored scientifically, clinically and from a regulatory perspective.

## 4. Facilitating Regulatory and Ethics Capacity for EM Trials

It is largely recognized that clinical research, especially HIV prevention research, should be conducted in countries and settings most affected by the HIV epidemic. This may include countries not accustomed to engaging in early-phase clinical research. EM trials may present new challenges for various stakeholders, including regulatory authorities, RECs and clinical trial sites, as discussed earlier.

Regarding *review by RECs*, investigators should draft protocols in a way that communicates the nature and merits of EM trials. REC reviewers may need opportunities to have their awareness of such studies raised. RECs may need to communicate and network more closely to harmonize their concerns. RECs should make extra efforts to avoid delays in approval (and researchers potential mitigate such delays by submitting well-crafted applications).

Regarding *review by NRAs*, several initiatives are currently working on capacity building in low- and middle-income countries. For example, the African Vaccine Regulatory Forum (AVAREF) represents the efforts of the African continent to support product development and ensure they meet the required standard for clinical testing. AVAREF has shown that convergence is possible, enabling sponsors to submit across the African continent. Funders such as the European and Developing Countries Clinical Trials Partnership (EDCTP) are also engaging with national regulators and AVAREF to support capacity building through an integrated approach (short- and long-term training) to ensure that well-trained and connected individuals can tap into resources available in other regions and countries with a commitment to equity.

Regarding review by *RECs and regulators*, some shared concerns were noted, including delays in the approval process, insufficient platforms for sharing ideas and communicating, and a lack of a robust electronic submission portal for clinical trial applications. Ways forward aim to address these issues by providing training opportunities, harmonizing, and synchronizing regulatory approval processes, building human resources, and developing technologies to support the review process. There is a recognized need for approaches that increase the quality of researcher submission to NRAs and RECs that shorten timeframes and increase the rigor of review. This will increase efficiency and support high-quality review while strengthening country collaboration and capacity building. African regulators have an important role to play and should take the lead and be supported.

## 5. Facilitating Coordination for EM Studies

It was noted that the need for better coordination of HIV vaccine research and development was recognized in 2003 [10] and led to the creation of organizations and structures to improve communication and coordination of studies and facilitate a more strategic approach to vaccine R&D. The field of EM for HIV vaccine development will benefit from such a shared platform that brings together major funders, researchers, regulators, and other relevant stakeholders.

Facilitation will be critical to ensure training of scientists at trial sites and motivate clinical research sites to take on EM studies, as well as engage participants to participate in this type of research. Ultimately, this may be more about focus and facilitation than coordination.

Supporting organizations, such as the Global HIV Vaccine Enterprise, have an important role to play as a neutral convenor to facilitate the coordination of the research. The Enterprise was established to unite stakeholders to share knowledge, foster collaboration, enable solutions, and expand support critical to accelerating the discovery and development of—and future access to—an HIV vaccine.

Open and transparent data sharing, as well as research activities, will eliminate duplication and facilitate communication of successes and failures. Validation of methods, experiments and assays and sharing of reagents, samples and data will allow for the validation and reproducibility of results.

## 6. The Future of EM for HIV Vaccine R&D

EM studies will require significant engagement of advocates and community stakeholders to elicit and respond to their needs and concerns. It is critical to nurture a body of advocates who understand the merits and challenges and can advise researchers and build literacy in community stakeholders. It was noted that the COVID-19 pandemic has brought back vaccines and their development into the public sphere, making us re-examine the traditional ways of developing vaccines.

The success of SARS-CoV-2 vaccine development has shown that it is possible to accelerate clinical research while maintaining safety standards with focused research steered by a sense of urgency. Once more, with another efficacy trial discontinued (HVTN 706/Mosaico [11]) and the next products to enter efficacy study at least a decade away, HIV vaccine development is at a crossroads and new approaches and renewed focus are required. EM can be a tool to take us closer to a safe and effective vaccine.

The Global HIV Vaccine Enterprise will work with partners in the field to continue advocate for research approaches, such as iterative experimental medicine that will enable and accelerate early clinical studies.

## Figures and Tables

**Table 1 vaccines-11-00970-t001:** Content of the experimental medicine webinars held in 2022.

Webinar 01—18 January 2022	
**What is experimental medicine and what are the key challenges?** Chair: Mark Feinberg, IAVI, USA	Key elements of experimental medicine research were discussed against the background of ethical challenges presented by this approach to research and development, with benefits that are not immediate to study participants. This webinar also highlighted how to engage communities to achieve support. https://youtu.be/RmRFO1vaBVM (accessed on 9 May 2023)
**Presentation**	
**Experimental Medicine Vaccine Trials EMVTs, Opportunities and Challenges.** Robin Shattock, Imperial College London, UK	
**Webinar 02—1 February 2022**	
**Experimental Medicine trials for HIV vaccine research** Chair: Nina Russell, the Bill & Melinda Gates Foundation, USA	This webinar looked into planned and ongoing experimental medicine trials for the development of HIV vaccines. It discussed strategy, products, and trial design to identify the most promising areas for development. https://youtu.be/qhuSamXNarI (accessed on 9 May 2023)
**Presentations**	
**An overview of the G001, G002, G003 and the HVTN study design.** William Schief, The Scripps Research Institute and International AIDS Vaccine Initiative, USA **Self-amplifying mRNA platform for COVID-19 vaccines.** Robin Shattock, Imperial College London, UK **VIR-1111 study—brief background about the CMV vector platform.** Ann Arvin, Vir Biotechnology, San Francisco, USA **The value of experimental medicine studies for vaccine development.** Brett Leav, Moderna, USA	
**Webinar 03—20 September 2022**	
**Building regulatory and ethics expertise for Experimental Medicine** Chair: Cathy Slack, University of KwaZulu-Natal, South Africa	This webinar explored the needs, optimal methods, and practical approaches for improving knowledge and capabilities of health products that regulatory authorities, agencies and Research Ethics Committees use to assess experimental medicine clinical research in high-income and low- and middle-income countries. https://youtu.be/sI8i1W_758w (accessed on 9 May 2023)
**Presentations**	
**Experimental Medicine trials: the IAVI G003 study.** Julien Nyombayire, Rwanda Zambia Health Research Group, Rwanda **Challenges of experimental medicine.** Jerome Singh, University of Toronto, Toronto, Canada **A perspective from a regional regulatory authority.** Eric Karikari-Boateng, Food and Drug Administration, Ghana	

**Table 2 vaccines-11-00970-t002:** Differences between experimental medicine trials and Phase I trials IND: investigational new drug, IMPD: investigational medicinal product dossier, IRB: Institutional Review Board, MCB: master cell bank.

	Traditional Phase I Trial	Experimental Medicine Trial
**Purpose of the trial**	Product development	Scientific information
**Next step**	Phase II (tentatively)	Improve vaccine design/Phase I
**Number of volunteers**	~20–100	Defined by scientific question
**Use of controls/placebo**	Yes	Potentially no
**Duration (months)**	~12–18	Usually < 12
**Laboratory monitoring of volunteers**	Safety Mostly regular immunogenicity	Safety Mostly special assays
**Preclinical (animal) evaluation**	Extensive (up to protection)	Limited/generic for platform (safety)
**Vaccine manufacturing**	Scalable product (reproducibility/MCB, etc.)	Pilot/small-scale lot
**Product characterization**	Suitable for Phase 3 trials; long-term stability	Description of product (qualified assays): purity, potency, stability
**Safety/toxicity**	Extensive	Limited
**Regulatory**	IND/IMPD	IND/IMPD
**Ethics**	IRB approval; involves large communities	IRB approval; involves individuals
**Industrial partner**	Highly desirable	Desirable, but not essential

## Data Availability

Not applicable.

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
