# Peer review of "Experimental Medicine for HIV Vaccine Research and Development"

_vaccines, 2023, doi:10.3390/vaccines11050970_

Round 1

Reviewer 1 Report

Review of “Experimental medicine for HIV vaccine research and development – meeting report” by Holly Pruden et al.

Despite more than 40 years of HIV vaccine research an HIV vaccine remains elusive. The failure of several candidate vaccines in recent large human clinical trials have prompted the HIV vaccine research and development (R&D) field to consider new approaches to developing vaccines. The manuscript summarizes the key questions and discussions at a series of virtual webinars between January to September 2022, hosted by the Global HIV Vaccine Enterprise at IAS – the International AIDS Society, about the merits and challenges of experimental medicine (EM) studies to accelerate the development of safe and effective HIV vaccines. The discussions involved scientists, policy makers, community stakeholders, advocates, bioethicists, and funders.

The meeting report is important and timely in providing information related to the merit, challenges, regulatory and ethics capacity, and coordination for EM studies for HIV vaccine in these discussions. The video links provided by the report are important and helpful for readers to listen to the discussions. 

Author Response

Reviewer 1
No comment to address.

Reviewer 2 Report

Given the failures of HIV vaccine trials and the slow pace of vaccine development, this summary of the experimental medicine webinars for the general public by Pruden and colleagues is great, timely and well written. It will be of great benefit to the readership of Vaccines and beyond. The writeup is really good. I have a few  suggestions: 

1. From their definition it is not clear to the reader if the candidate immunogens for EM would have already been tested in animals for safety data. This should be clarified in the introduction. 

2. Who attended these webinars? In other words whose opinions are being shared. It will be great for readers if the authors  show a map of where the webinar attendees came from? This will give confidence to readers that the ideas shared are broad based. 

3. Line 137 'considerations' rather than 'considered'

Author Response

Reviewer 2
1. Clarify whether the candidate immunogens for EM would have already been tested in animals for safety data.
Table 2 and the introduction (lines 86 and 114), indicate that limited, or platform-specific, pre-clinical evaluation would be performed for products prior to testing in EM trials. This would include safety (toxicity study) as conducted for any product prior to use in human clinical trials. A sentence was added to clarify (line 115).

2. Who attended these webinars?
The report provides a comprehensive summary of the panel discussions held during each event. The Acknowledgement section (line 275) includes a list of all participants, carefully selected to ensure diverse representation across geographical locations, genders, and professional constituencies. While we recognize the importance of visual aids, we believe that including a map would not enhance the report's clarity or effectiveness.

3. Line 137 'considerations' rather than 'considered'

Change made.

Reviewer 3 Report

This paper is a review on a series of meetings discussing the idea of using the experimental medicine approach for HIV vaccine research and development. The authors explain the merits of experimental medicine and point out the differences between experimental medicine trials and Phase 1 trials. The authors also discuss the challenges of conducting experimental medicine trials and more importantly the regulations and ethics of conducting such trials. This was a good introduction to the experimental medicine approach for HIV vaccine research, as well as a review on what should be considered and discussed when planning and coordinating such a trial. 

There are several points that I think could be addressed:

- I think that there are several Phase 1 trials that are taking the experimental medicine approach. NCT05423418 and NCT04658667 are two Phase 1 trials that seem to take the experimental medicine approach (if my interpretation of the EM approach is correct). 

- The authors should reconsider Table 2 in stating that the Experimental Medicine trial would potentially not use controls or placebos. Any work that is asking a scientific question (which would one of the points of experimental medicine) should have a control or placebo group. 

Author Response

Reviewer 3
1. There are several Phase 1 trials that are taking the experimental medicine approach. NCT05423418 and NCT04658667 are two Phase 1 trials that seem to take the experimental medicine approach.
This is correct. No comment to address.

2. The authors should reconsider Table 2 in stating that the Experimental Medicine trial would potentially not use controls or placebos. Any work that is asking a scientific question (which would one of the points of experimental medicine) should have a control or placebo group.
Table 2 highlights that EM trials are distinct from other clinical trials in that they do not always require a control or placebo group. The primary objective of these trials is to answer a specific research question, often related to the immune response triggered by a vaccine, to accelerate the development of more promising vaccine candidates. One way EM trials can achieve this is by evaluating the immune response against an immunogen compared to a pre-immunization baseline, or by comparing the immune response to various related immunogens to identify the most promising candidates to move forward with. These trials typically involve a small number of participants and use well-characterized vaccine platforms and laboratory assays to measure immune responses. In the case of HIV research, for example, pre-vaccination levels serve as a reliable control for evaluating the immune response, making the use of a placebo unnecessary. The use of a placebo would add complexity, increase costs, and raise ethical concerns. For example, the reviewer identified trial NCT05423418 which does not include a placebo or control but assesses different biologicals head-to-head to optimize the dosing of a vaccine adjuvant.

Reviewer 4 Report

Thank you for the opportunity to review this article.

I think this is an opinion paper but I find it hard  to consider it an article of scientific relevance. I suggest to reconsider submission as opinion paper. How a meeting reports can draw the attention of the scientific communty?

Please, I ask the authors to better specify this point.

  ask the authors to better specify this point

Author Response

  1. Suggest reconsidering submission as an opinion paper and query how a meeting report can draw the attention of the scientific community?
    This manuscript serves as a report on three virtual events and summarizes the expert opinions shared during these events. As a meeting or conference report published in Vaccines, it provides valuable insights into the latest developments and perspectives in the field. Nevertheless, we defer to the Editor for final judgment on the classification of this manuscript. The development of an HIV vaccine is facing increasing challenges due to the widespread availability of effective antiretroviral-based HIV prevention measures. The failure of three large-scale trials to demonstrate efficacy has necessitated a return to the discovery and preclinical stages of HIV vaccine development. Therefore, it is imperative to explore novel approaches to research that allow for rapid iteration during the early stages of clinical trials. This approach represents a significant departure from traditional product development paths, which emphasize long-term development from inception to completion. The use of EM trials is a promising method to identify the next generation of HIV vaccine products, but several challenges must be addressed. This series of events aims to draw the attention of product developers worldwide to this innovative approach, and thus facilitate progress towards the ultimate goal of an HIV vaccine.

Round 2

Reviewer 4 Report

I'm sorry, but although the authors have replied fervently explaining the intentions of the manuscript, in its current form I do not recommend the publication.